# Neutrophil Diversity (Immature, Aged, and Low-Density Neutrophils) and Functional Plasticity: Possible Impacts of Iron Overload in β-Thalassemia

**DOI:** 10.3390/ijms251910651

**Published:** 2024-10-03

**Authors:** Kritsanawan Sae-Khow, Awirut Charoensappakit, Asada Leelahavanichkul

**Affiliations:** 1Department of Microbiology, Faculty of Medicine, Chulalongkorn University, Bangkok 10330, Thailand; kritsanawan_29@hotmail.com (K.S.-K.); awirut.turk@gmail.com (A.C.); 2Center of Excellence in Translational Research in Inflammation and Immunology (CETRII), Faculty of Medicine, Chulalongkorn University, Bangkok 10330, Thailand; 3Division of Nephrology, Department of Medicine, Faculty of Medicine, Chulalongkorn University, Bangkok 10330, Thailand

**Keywords:** neutrophil diversity, thalassemia, low-density neutrophils, iron overload

## Abstract

Neutrophil dysfunction is a form of immune suppression in patients with β-thalassemia (Beta-thal), although data on this are limited. In this study, blood from patients and healthy volunteers was analyzed. Flow cytometry analysis demonstrated an increase in immature neutrophils (CD16− CD62L+) and aged (senescent) neutrophils (CD16+ CD62L−) in Beta-thal patients compared to healthy volunteers. The Beta-thal neutrophils demonstrated less prominent chemotaxis and phagocytosis than healthy neutrophils at the baseline. With phorbol myristate acetate (PMA) or lipopolysaccharide (LPS) stimulations, some of the indicators, including the flow cytometry markers (CD11b, CD62L, CD66b, CD63, apoptosis, and reactive oxygen species) and neutrophil extracellular traps (NETs; detected by anti-citrullinated histone 3 immunofluorescence), were lower than the control. Additionally, low-density neutrophils (LDNs), which are found in the peripheral blood mononuclear cell (PBMC) fraction, were observed in Beta-thal patients but not in the control group. The expression of CD11b, CD66b, CD63, arginase I, and ROS in LDNs was higher than the regular normal-density neutrophils (NDNs). The proliferation rate of CD3+ T cells isolated from the PBMC fraction of healthy volunteers was higher than that of the cells from patients with Beta-thal. The incubation of red blood cell (RBC) lysate plus ferric ions with healthy NDNs transformed the NDNs into the aged neutrophils (decreased CD62L) and LDNs. In conclusion, iron overload induces neutrophil diversity along with some dysfunctions.

## 1. Introduction

Beta-globin (β-globin) is a subunit of hemoglobin, an iron-containing protein used for oxygen transport in red blood cells (RBCs), where it is located along with alpha-globin [1]. β-thalassemia is chronic hemolytic anemia due to mutations in the β-globin gene, causing a reduction or an absence of β-globin synthesis, which might require regular blood transfusions in the most severe form of the disease [1]. Characteristics of the peripheral blood smear of an individual with thalassemia show hypochromic microcytic RBCs with anisopoikilocytosis (varying sizes and shapes of RBCs in the blood smear), which are able to carry less oxygen than normal RBCs [1,2]. Several studies on β-thalassemia have demonstrated defects in the immune system of patients with β-thalassemia as an important factor contributing to poor outcomes after infections, which is the most common cause of mortality and morbidity, particularly in patients with transfusion-dependent β-thalassemia (TDβT) [3,4]. Indeed, the most common cause of infection in patients with β-thalassemia is pneumonia, which is followed by infective diarrhea [3,4]. Although the abnormal immune responses in patients with β-thalassemia are partly related to the well-known clinical aspects of the disease, such as hemochromatosis (severe iron overload with iron deposition in the internal organs, especially liver cirrhosis, cardiomyopathy, and endocrinopathy) and splenectomy (the removal of the spleen due to the accelerated destruction of donor RBCs in the spleen or complications from severe splenomegaly in the abdomen), increased susceptibility to infection in patients without either severe hemochromatosis or splenectomy has also been observed [3,4]. Moreover, the increased susceptibility to infection and abnormalities in the immune responses of patients with TDβT might also be due to the translocation of microbial molecules or viable organisms from the gut into the blood circulation, referred to as leaky gut, leading to the presence of bacterial lipopolysaccharide (LPS; a major component of the cell wall of Gram-negative bacteria) in the blood (endotoxemia) with or without bacteremia [5,6]. Despite the uncertain mechanism behind thalassemia-induced leaky gut or spontaneous endotoxemia (the presence of LPS in the blood circulation without bacteremia or active infection), iron overload in enterocytes might be partially contributing to this phenomenon [5,6]. While gut-derived bacteremia may lead to direct infection in several internal organs (such as liver abscess, endocarditis, and sepsis), the chronic exposure of LPS to the immune system can also induce immune exhaustion, which might increase susceptibility to infection and sepsis in patients [5,6]. Because hemochromatosis is a major side effect of frequent blood transfusion in these patients and impacts of iron overload against immune cells, for example, monocytes and macrophages, are mentioned [5,6], iron chelation (the binding of iron with high-affinity molecules to remove the excess iron in the body) is a standard treatment to reduce the severity of iron deposition in several internal organs that might also attenuate iron overload-induced immune dysfunction [1]. The impacts of iron chelators in the attenuation of hemochromatosis of the internal organs are well known, but these molecules might also indirectly improve the immune response of patients through the attenuation of iron toxicity against immune cells.

Hence, there are extrinsic factors (blood transfusion and splenectomy) and intrinsic components (hemochromatosis and leaky gut) that can increase the risk of infection in patients with β-thalassemia, and a better understanding of immune defects in these patients may be clinically useful. In general, innate immunity is a system that responds to infection mainly to rapidly limit microbial burdens using phagocytic cells (neutrophils, macrophages, and dendritic cells) and degranulation cells (basophils, eosinophils, and mast cells), while adaptive immunity (T cells and B cells) is responsible for the eradication of the pathogenic organisms [7,8]. Indeed, several studies have demonstrated that β-thalassemia can be attributed to the coexistent defects in both the innate and adaptive immune systems [7,8], for example, reduced levels of complement components, decreased T cell proliferation, and impaired B cell differentiation with poor antibody production [8,9,10]. Even though neutrophils, the largest population of circulating white blood cells in humans, are known as the first defense against infection [11], data on neutrophil functions and the kinetics of neutrophils in β-thalassemic conditions are still limited. As such, neutrophils are the first line of defense against infections, especially bacteria, via several functions, including chemotaxis, phagocytosis, oxidative burst, and neutrophil extracellular traps (NETs), and they can initiate inflammation and control microbial burdens [11]. Although neutrophils have long been considered a relatively homogeneous terminally differentiated population, evidence of neutrophil heterogeneity is currently emerging [12,13]. Data from the past decade have revealed that neutrophils may also exert immunoregulatory functions and display phenotypic changes in tissue microenvironments, demonstrating functional plasticity. Indeed, heterogeneous populations of circulating neutrophils have been described based on discrete parameters both in healthy and in pathological conditions (cancers, infections, and autoimmune diseases) [12,13,14], and the traditional belief that neutrophils are a uniform population of short-lived cells has been challenged. Despite several reports on thalassemia-induced immune dysfunction, data on neutrophil characterization in β-thalassemia are still limited. Thus, this study not only aimed to determine neutrophil diversity but also sought to evaluate neutrophil functions in patients with β-thalassemia. Here, neutrophils from patients with β-thalassemia major were explored using flow cytometry analysis with ex vivo tests by several inducers.

## 2. Results

### 2.1. Neutrophil Diversity in β-Thalassemia

Table 1 presents the epidemiologic data of patients and healthy volunteers (controls). Flow cytometry analysis from the neutrophil fraction after gradient separation demonstrated more than 95% CD66b-positive cells (total neutrophils) before further determination (Figure 1A). Then, neutrophils were categorized by FcγRIII (CD16) and L-selectin (CD62L) into (i) immature neutrophils, CD16 negative with CD62L positive (CD16− CD62L+), (ii) mature neutrophils (CD16+ CD62L+), and (iii) aged neutrophils (CD16+ CD62L-) following previous publications [15,16]. Notably, CD184 (C-X-C chemokine receptor type 4 or CXCR4), an adhesion molecule [15,16], was additionally tested in aged neutrophils (CD66b+ CD16+ CD62L−) (Figure 1A, right). An abundance of immature and aged neutrophils in patients with β-thalassemia was higher than healthy controls as evaluated by flow cytometry analysis (Figure 1B) but was not different by conventional cell morphology using Wright’s stain (Figure 1C). For neutrophil functions, phorbol myristate acetate (PMA) or lipopolysaccharide (LPS), a potent neutrophil stimulator and strong pro-inflammatory activator, respectively, were incubated with isolated neutrophils (gradient separation) before evaluation by flow cytometry to determine (i) cell activation using anti-CD11b (β-integrin) and anti-CD62L (L-selectin), and (ii) cell secretory function using anti-CD66b (carcinoembryonic antigen-related cell adhesion molecule 8 or CEACAM8) and anti-CD63 (Tetraspanin) (Figure 1D–H).

In neutrophils from healthy controls, PMA elevated all of these markers, while LPS enhanced nearly all markers except for CD63 (Figure 1E–H). The responses of neutrophils from patients with β-thalassemia against PMA were similar to control neutrophils, while LPS elevated only CD11b and CD62L but not CD66b and CD63 when compared with controls (Figure 1E–H). The stimulated neutrophils (by PMA or LPS) demonstrated an upregulation of most of these biomarkers when compared with unstimulated neutrophils; however, characteristics of the stimulated neutrophils from patients and from volunteers were similar (Figure 1E–H). Notably, LPS induced a lower expression of CD11b and CD66b in neutrophils from patients than volunteers (Figure 1G,H), which implied a lower neutrophil activity in β-thalassemia compared with the cells from healthy controls [17].

### 2.2. Defects of Neutrophil Function in β-Thalassemia

The mechanisms of neutrophils to protect against infection are chemotaxis, phagocytosis, reactive oxygen species (ROS) production (oxidative burst), and the generation of neutrophil extracellular traps (NETs; the web-like structure primarily produced by DNA) [18]. At the resting state (unstimulated cells), neutrophils from patients with β-thalassemia exhibited less effective chemotaxis and phagocytic capabilities (Figure 2A,B), with a similarity in ROS production, NET formation, and apoptosis level when compared with neutrophils from healthy controls (Figure 2C–F). With PMA, all of these parameters, except for ROS production and NETs, were similar between patients with β-thalassemia and healthy controls (Figure 2A–F). Notably, ROS and NETs in PMA-stimulated neutrophils from patients with β-thalassemia were lower than the activation in healthy controls (Figure 2C–E). Meanwhile, LPS elevated all of these neutrophil functions at a level similar to PMA stimulation; however, there was less NET formation and cell apoptosis in β-thalassemic neutrophils when compared with the stimulated neutrophils from volunteers (Figure 2D–F). On the other hand, calcium ionophore, another NET stimulator, also similarly induced NET formation in β-thalassemic neutrophils and control neutrophils (Figure 2E), while calcium ionophore did not alter other neutrophil functions when compared with unstimulated neutrophils from healthy controls.

Because of more prominent immature and aged neutrophils in patients with β-thalassemia compared with healthy controls (Figure 1B), the lower abundance of regular neutrophils in patients might be responsible for defects of neutrophil functions. Then, the correlation between the percentage of immature and/or aged neutrophils and these functional parameters was calculated by Spearman’s rank correlation, and the r-square (r^2^) values (coefficient of determination) were demonstrated through a correlation heat map picture (Figure 2G). Overall, immature (CD16− CD62L+) and aged (CD16+ CD62L−) neutrophils showed a trend toward negative correlation in most of the functional parameters (most of the heat map shows a blue color demonstrating a lower r^2^), and only a few pairs of correlation reached significant values (Figure 2G). As such, a positive correlation (red color) was demonstrated only in (i) the chemotaxis of immature neutrophils (unstimulating and PMA stimulation), (ii) ROS and NETs of aged unstimulated neutrophils, and (iii) the apoptosis of aged LPS-stimulated neutrophils; however, these parameters did not reach statistical significance (Figure 2G). Meanwhile, the significant negative correlations were (i) phagocytosis of unstimulated neutrophils (immature and aged cells), (ii) phagocytosis and NETs of PMA-stimulated neutrophils (immature and aged cells), and (iii) phagocytosis, NETs, and apoptosis of LPS-stimulated neutrophils (immature and aged cells) (Figure 2G). These findings implied that neutrophil diversity (increased immature and aged neutrophils) in β-thalassemia may be a potential cause of dysfunctional activities.

### 2.3. Elevation of Low-Density Neutrophils and the T Cells Suppression in β-Thalassemia

Recently, low-density neutrophils (LDNs), the neutrophils identified in the PBMC fraction after density gradient separation, are demonstrated as a unique neutrophil population and mostly reported as myeloid-derived immune suppressor cells in several diseases, especially cancers [19]. In addition to the expanded subsets of regular neutrophils (normal-density neutrophils, NDNs) in patients with β-thalassemia, LDNs might be another interesting subset. As such, a representative picture of PBMCs and neutrophil fractions using density gradient centrifugation is demonstrated in Figure 3A. Then, flow cytometry gating from the PBMC fraction using CD66b (a marker for neutrophils) (Figure 3B) demonstrated more prominent LDNs (CD66b-positive cells in the PBMC fraction) in patients with β-thalassemia and was rarely observed in healthy controls (Figure 3C). Differences in the morphology between LDNs and normal density neutrophils (NDNs) could not be observed through Wright’s stain using a light microscope because LDNs in either patients or healthy controls consisted of segmented mature neutrophils and immature cells (band form) (Figure 3D). In comparison with regular neutrophils (NDNs) from patients (Beta-thal NDNs) or healthy controls (healthy NDNs), LDNs from β-thalassemia (Beta-thal LDNs) exhibited higher CD11b expression with lower CD62L expression as well as elevated degranulation markers (CD66b and CD63) (Figure 3E–H). Notably, the immune suppressive properties of LDNs have been previously demonstrated through increased ROS production, elevated PD-L1 expression, and secretion of the arginase I enzyme to deplete L-arginine (an essential amino acid required for normal T cell development) from T cells [20,21]. Here, unstimulated Beta-thal LDNs demonstrated higher ROS generation than NDNs (both from patients and volunteers), while PMA elevated ROS in all neutrophil subgroups (Figure 3I). With PMA, increased ROS production in healthy NDNs was similar to the levels of Beta-thal LDNs, while the ROS of Beta-thal NDNs was at the lowest level (Figure 3I, right). On the other hand, PD-L1 expression was not different between NDNs (both from patients and volunteers) and Beta-thal LDNs (Figure 3J). In parallel, Beta-thal LDNs produced more arginase I than NDNs (both from patients and volunteers) (Figure 3K), which possibly correlated with higher plasma arginase I levels in patients than volunteers (Figure 3L). There was a positive correlation between plasma arginase I and the percentage of LDNs in patients with β-thalassemia (Figure 3M), implying the possible interference on T cell functions and the use of plasma arginase I as a biomarker [22,23].

For T cells, there was no difference in the lymphocyte count and percentage of CD4+ and CD8+ T cells between β-thalassemia and healthy controls (Figure 4A,B). Because the PBMC fraction contains more condensed immune cells (monocytes, lymphocytes, and LDNs) than peripheral blood, the T cell activities from these parts might be different and were separately investigated. Notably, CD3+ T cells isolated from peripheral blood or PBMC were incubated with T cell stimulators (anti-CD3 and anti-CD28 dyna-bead) for 4 days before measurement of T cell proliferation using carboxyfluorescein succinimidyl ester (CFSE) (Figure 4C). As such, the proliferation rate of T cells from peripheral blood (isolated T cells) was not different from the PBMC fraction (PBMC T cells) (Figure 4D, left) in healthy controls, while peripheral blood T cells more prominently proliferated than PBMC T cells in patients (Figure 4D, right). Notably, the proliferation rate of peripheral blood T cells and PBMC T cells from patients was lower than the cells from volunteers (Figure 4D, between left and right). These data implied different T cell functions in the peripheral blood and in the PBMC fraction in patients with β-thalassemia.

### 2.4. Impacts of Hemolysis and Iron Overload in Neutrophil Functions, the In Vitro Experiments

Chronic hemolysis and iron overload are common pathophysiologies that are correlated with the disease severity of β-thalassemia [1] and might affect neutrophil functions. Then, isolated neutrophils, using density gradient separation, from healthy volunteers were treated with RBC lysate using RBCs from normal volunteers with and without ferric ions (see method), representing neutrophils with hemochromatosis and iron overload. As such, a combination of RBC lysate and ferric ions (RBC lysate + Fe^3+^) and Fe^3+^ alone, but not RBC lysate alone, differentiated mature neutrophils (CD16+ CD62L+) into aged neutrophils (CD16+ CD62L−) without any difference in the abundance of immature-liked neutrophils (CD16− CD62L+) (Figure 5A,B). In parallel, RBC lysate + Fe^3+^ and Fe^3+^ alone, but not RBC lysate alone, also generated LDN populations from regular normal-density neutrophils (NDNs) (Figure 5C,D). Because of the more prominent effect of RBC lysate + Fe^3+^ over RBC lysate alone in the induction of the neutrophil subpopulation (Figure 5A–D), only RBC lysate + Fe^3+^ was further used to stimulate NDNs and LDNs. As such, RBC lysate + Fe^3+^ more prominently elevated CD11b (an activation marker) and CD66b (a secretory marker) in Beta-thal neutrophils (NDNs and LDNs) without CD63 upregulation (Figure 5E–H). Meanwhile, the chemotaxis of RBC lysate + Fe^3+^-activated LDNs, but not in activated NDNs, was lower than untreated NDNs (Figure 5H). For NET formation, there was no difference in NETs between untreated NDNs and stimulated neutrophils (NDNs and LDNs), while PMA induced higher NETs than stimulated neutrophils (NDNs and LDNs) (similar NETs between NDNs and LDNs after stimulation) (Figure 5I). LPSs similarly induced NETs in unstimulated NDNs and RBC lysate + Fe^3+^-activated NDNs, which were higher than RBC lysate + Fe^3+^-activated LDNs (Figure 5I). Interestingly, LDNs from patients and LDNs from the in vitro induction using RBC lysate + Fe^3+^ (or Fe^3+^ alone) were different. As such, the LDNs from blood consisted of three subpopulations, including immature neutrophils (CD16 CD62L+), mature neutrophils (CD16+ CD62L+), and aged neutrophils (CD16+ CD62L−) (Figure 6A, upper), while the induced LDNs consisted of only CD16+ CD62L+ cells with negative CD184 (Figure 6B, upper). Notably, CD184, or C-X-C chemokine receptor type 4 (CXCR4), is a chemokine with potent chemotactic activity, mostly on lymphocytes, and it is expressed only in a few neutrophil subpopulations, possibly for the shifting of neutrophils from the blood into the bone marrow [15]. Then, CD184 is frequently used for distinguishing neutrophil characteristics. Here, CD184 was expressed only in blood-isolated LDNs with an immature characteristic (CD16− CD62L+) and a senescent feature (CD16− CD62L+), while CD184 was negative in blood-isolated LDNs with a maturity biomarker (CD16+ CD62L+) and in inducing LDNs (Figure 6A,B, lower). Despite differences between peripheral blood LDNs and induced LDNs, a proof of the concept on impacts of iron overload is demonstrated. These findings suggest that hemolysis and iron overload in patients with β-thalassemia are among several factors inducing neutrophil subpopulations (neutrophil diversity) and might alter neutrophil functions, leading to increased susceptibility to infection.

## 3. Discussion

Neutrophils are one of the important immune cells for microbial control through several functions, including chemotaxis, phagocytosis, oxidative burst, and the formation of NETs [24,25]. The dysfunction of neutrophils leads to systemic infection, angiogenesis impairment, and delayed wound healing [18]. Although functional defects of neutrophils in β-thalassemia are partly due to an increase in immature neutrophils (band form), the chemotaxis and phagocytotic functions of the band form are comparable to those of mature neutrophils [16]. Then, increased immature neutrophils do not fully explain the elevated susceptibility to infection in patients with β-thalassemia (Beta-thal). Due to limitations of neutrophil evaluation by morphological features (multi-segmented mature neutrophils and immature band form), flow cytometry analysis based on Fc gamma receptor (FcgR) III (CD16; a receptor for the Fc region of gamma immunoglobulins) and L-selectin (CD62L; an adhesion molecule) categorized neutrophils into immature cells (CD16− and CD62L+), aged neutrophils (CD16+ and CD62L−), and mature neutrophils (CD16+ and CD62L+) was used [16,26]. 

In comparison with mature neutrophils, immature neutrophils (CD16 negative cells) show a trend toward a lower antibacterial capability, which was partly due to the less effective intracellular acidification [27,28]. Meanwhile, aged neutrophils (CD62L negative cells) demonstrate a chemotaxis defect because of a decrease in L-selectin (an adhesion molecule) with impaired extravasation capability [27]. Here, the higher abundance of immature and aged neutrophils in patients than healthy volunteers correlated with chemotaxis and phagocytosis defects without impairment on NETs, ROS production, and apoptosis (unstimulated condition). Notably, reduced phagocytic activity against fluorescent-stained bacteria of neutrophils from patients compared with volunteers (Figure 2B) using the pHrodo^TM^
*S. aureus* Bioparticle^TM^ (Thermo Fisher) implied a less effective bacterial control. Despite more prominent iron levels in the blood of patients than volunteers (high serum ferritin in Table 1), there was no difference in the apoptotic cell death of neutrophils in patients compared with volunteers (Figure 2F). Although the profound endocytosis of free iron and ferritin by neutrophils and monocytes/macrophages induced iron-associated cell toxicities [29,30], neutrophil apoptosis was not different between patients and volunteers (Figure 2C). Similarly, there is no macrophage cell death even with the incubation with a very high dose of ferric ions (800 µM) in a previous publication [6]. Perhaps both neutrophils and macrophages can tolerate some ranges of iron overload, or the injured neutrophils/macrophages might be found only in the blood that is trapped inside the spleen (splenic sequestration). With PMA, a potent activator of protein kinase C (PKC), there were lower NETs and ROS in neutrophils from patients, implying defects on PKC and microbial control [11,27] in β-thalassemia. Likewise, LPS reduced CD11b (integrin alpha M for extravasation) in neutrophils from patients (Figure 1E–H and Figure 2A–F), supporting various microbicidal defects in β-thalassemia [9,11,31]. More studies on these topics are interesting. 

Not only immature and aged neutrophils, but low-density neutrophils (LDNs) in the PBMC fraction (consisting of mature multi-segmented cells and immature band form) (Figure 3C,D) were also more prominent in patients than volunteers. In comparison with healthy normal-density neutrophils (NDNs), LDNs demonstrated more prominent functions in (i) secretory property (high CD66b and CD63), (ii) ROS production, and (iii) T-cell suppression (high arginase 1) with a cell activity balance (high CD11b but low CD62L). Thus, the increased degranulation (upregulated CD66b and CD63) might reduce cell density and alter NDNs into LDNs. Also, an upregulated *arginase I* in thalassemic LDNs was similar to LDNs reported from other conditions (non-thalassemic hemoglobinopathies and cancers) with T cell suppression property [19,21,32]. Despite a non-direct LDN and T cell incubation due to the restricted blood volume of collection in our study, the lower proliferation rate of T cells from patients (from PBMCs and peripheral blood) than volunteers supported T cell dysfunction in β-thalassemia. Although T cell dysfunction in β-thalassemia [23,33,34] and T cell suppression by LDNs [19,21,32] in non-thalassemic conditions (cancers, chronic inflammation, pregnancy, and infancy) [35,36] are mentioned, data on the roles of LDNs in T cells of patients with β-thalassemia are still very limited. While LDNs in cancer suppress T cells via the PD-L1:PD-1 axis [37,38], arginase I in β-thalassemia LDNs might hydrolyze L-arginine, which is an essential amino acid for T cell receptors [39,40], as observed by an increase in plasma arginase I (Figure 3L) similar to other hemoglobinopathies [41]. Likewise, improved T cell proliferation by arginase inhibitors in cancers also supports the roles of arginase I in T cell suppression [42,43]. More sophisticated experiments are required for a solid conclusion on LDN-induced T cell dysfunction in β-thalassemia. Because intracellular iron homeostasis or iron chelation interfere with T cell functions [44], immunological defects in most of our patients might partly because of high ferritin and iron chelator administration. Despite the lack of a direct experiment on LDN-T cell interaction, the increased susceptibility to infection in patients with β-thalassemia might be due to (i) an increase in immature and aged neutrophils with defects on chemotaxis and phagocytosis, (ii) an elevation of LDNs with a possible myeloid-derived suppressor cell activity (prominent *arginase I*), and (iii) abnormal T cell proliferation.

Interestingly, red blood cell lysate with ferric ions (RBC lysate + Fe^3+^) or Fe^3+^ but not RBC lysate alone altered regular neutrophils into aged cells and LDNs, highlighting iron overload-induced neutrophil dysfunction and supporting the use of iron chelation to prevent neutrophil dysfunction in β-thalassemia. Although the density of LDNs from peripheral blood was similar to the LDNs from in vitro iron overload induction, the expression of CD16, CD62L, and CD184 was different. While the peripheral blood LDNs could be separated into mature, immature, and aged neutrophils with CD184 positive in some populations, the induced LDNs expressed only markers of mature neutrophils (CD16+ CD62L+) that were negative for CD184. To make the properties of in vitro LDNs closer to those of blood-derived LDNs, additional factors are needed. Despite differences between blood-derived LDNs and in vitro LDNs, impacts of iron overload using RBC lysate + Fe^3+^ on LDNs differed from those of NDNs. Although both RBC lysate + Fe^3+^ and Fe^3+^ altered neutrophils, only the former factor was used because it resembled patients’ hemolysis with iron overload. Indeed, RBC lysate + Fe^3+^-stimulated LDNs expressed higher CD11b and CD66b (activity markers) with lower chemotaxis and NETs compared with activated NDNs. Thus, iron overload causes neutrophil diversity and immune dysfunction (induction of aged cells and LDNs), and the use of iron chelation in patients with β-thalassemia is supported. Moreover, downregulated PU.1 causes neutrophil maturation defects in mice and humans with β-thalassemia [25]. Meanwhile, ineffective erythropoiesis and chronic inflammation in β-thalassemia induce erythroprogenitor cell death, bone marrow microenvironmental alteration [45,46], and neutrophil diversity interference (an altered balance between aged circulating neutrophils and newly released cells). In the blood circulation of patients, aged neutrophils express high CXCR4, possibly to stay inside the hematopoietic stem cell niche [15,26], and bone marrow alteration might affect regular processes of immune cells. More studies on thalassemia-induced immune suppression are still interesting.

Several limitations should be mentioned. First, the conclusion was based on a single-center analysis with a limited number of patients. The multicenter, larger cohorts with patients in different stages of β-thalassemia and various treatments might produce more varied results. Second, more sophisticated and detailed mechanistic experiments, such as proteomic or transcriptomic tests between LDNs and NDNs, are required for a solid conclusion on neutrophil diversity. Third, CD16 (FcγRIII) and CD62L [13,28,47], but not CD15 (Lewis x) and CD32, were used for neutrophil categorization with limited molecular indicators for NDN–LDN differentiation. Notably, many indicators, for example CD66b and CD11b, presented in both NDNs and LDNs [14,48]. There are three FcγRs in human neutrophils. As such, FcγRIII (CD16) and FcγRII (CD32) are neutrophil activity biomarkers due to the recognition of immune complexes [49], FcγRI (CD64) is an inflammatory marker from the binding to monomeric IgG during active inflammation [50], and CD32 is frequently mentioned in the diseases with circulating immune complexes [51,52]. More tests on CD32 and CD15 in NDNs and LDNs in patients with β-thalassemia will be very interesting. Despite several limitations, our data provide proof of the concept that iron overload might be an important factor inducing several subpopulations of neutrophils (neutrophil diversity) and interfere with several neutrophil functions.

## 4. Materials and Methods

### 4.1. Participants

Patients with β-thalassemia from the Thalassemia Unit of the King Chulalongkorn Memorial Hospital (KCMH) were enrolled according to the approved protocol (IRB No. 0481/66) from the Ethics Committees of the KCMH to use blood samples with written informed consent according to the guideline. The diagnosis of thalassemia was made through screening tests and the molecular characterization of globin mutations. The study included 23 patients with β-thalassemia major and 20 healthy volunteers as epidemiologic data in Table 1. All patients regularly received blood transfusions to maintain the hemoglobin levels between 9 and 10.5 g/dL.

### 4.2. Isolation of Peripheral Blood Mononuclear Cells (PBMC) and Neutrophils

Heparinized blood samples were collected and were processed within 2 h after the collection. Neutrophils were isolated from whole blood using double-layered density gradient centrifugation containing Ficoll-Paque (upper part) and PolymorphPrep^®^ (lower part) at a ratio of 1:1:1 before the centrifugation at 800× *g* for 30 min at room temperature without deceleration. The peripheral blood mononuclear cells (PBMCs) fraction, containing low-density neutrophils (LDNs), monocytes, and T cells, were at the interface between plasma and Ficoll-Paque layers, while regular neutrophils, referred to as normal density neutrophils (NDNs), were retrieved from the interface between Lymphoprep^®^ (Stem cell Technologies, Vancouver, BC, Canada) and PolymorphPrep^®^ (Serumwerk, Bernburg, Sachsen-Anhalt, Germany). The isolated PBMC and NDN fractions were washed in 1× deionized phosphate buffer solution (D-PBS) and eradicated contaminated red blood cells (RBCs) by RBC lysis buffer (BioLegend, San Diego, CA, USA) before resuspension in the staining buffer (2% fetal bovine serum with 0.1% sodium azide in 1× D-PBS) or the complete RPMI-based media. The cell viability was observed by Trypan blue dye staining (Thermo Fisher, Waltham, MA, USA).

### 4.3. Neutrophil Stimulation

The isolated neutrophils from gradient separation, as mentioned above, were incubated with RPMI1640 culture media (Gibco, Carlsbad, CA, USA) containing 10% heat-inactivated fetal bovine serum (FBS) (Gibco, Carlsbad, CA, USA). To test neutrophil properties, neutrophils from healthy volunteers at 2.5 × 10^5^ cells were stimulated for 2 h at 37 °C in 5%CO_2_ with several molecules, including (i) the potent inducer for neutrophil extracellular traps (NETs) using phorbol 12-myristate 13-acetate (PMA) (Sigma-Aldrich, St. Louis, MO, USA) at 100 ng/mL [51] or calcium ionophore A23187 (Santa Cruz Biotechnology, Dallas, TX, USA) at 100 nM/mL [53], (ii) the representative potent pro-inflammatory stimulator through lipopolysaccharide (LPS) using LPS from *Escherichia coli* 026:B6 (Sigma-Aldrich, St. Louis, MO, USA) at 100 ng/mL [54,55], or (iii) the representative iron overload using the lysate of RBCs from healthy volunteers (RBC lysate) alone or RBC lysate plus ferric chloride (FeCl_3_) (Sigma-Aldrich, St. Louis, MO, USA) at 400 μM/well [5,6]. For RBC lysate preparation, RBCs were lysed with distilled water (DI) at 80% hematocrit (HCT) in liquid nitrogen for 15 min. After complete lysis, the sample was centrifuged at 1000× *g* for 10 min to remove RBC debris, hemoglobin (Hb) in the supernatant was measured by Drabkin’s solution method, and RBC lysate at 500 ng/mL of Hb with or without ferric chloride (Sigma-Aldrich, St. Louis, MO, USA) using ferric ions at 250 uM/well was used. Then, cell activities were analyzed by flow cytometry (mentioned later). For low-density neutrophils (LDNs) in vitro induction, the neutrophils after 2 h activation by RBC lysate or ferric ions or both factors combined were centrifuged in density gradient separation before the retrieval of NDNs and LDNs from the upper and lower parts of the Lymphoprep^®^/Polymorprep, respectively. For chemotaxis, a previously published protocol [5,6] was used. Briefly, the stimulated neutrophils at 1 × 10^6^ cells/mL were added into the hanging inserts of the 24-well culture plates coated with N-Formylmethionyl-leucyl-phenylalanine (fMLP; a chemoattractant) (Sigma-Aldrich, St. Louis, MO, USA) with a 3 µm pore-size filter and further incubated for 90 min at 37 °C in a 5% CO_2_ incubator before the detection of migrated neutrophils using trypan blue staining. In parallel, neutrophil extracellular traps (NETs) were measured by immunofluorescence following a previous protocol [56,57]. Briefly, sterile coverslips were placed into a 24-well plate, and neutrophils at 2.5 × 10^5^ cells were inoculated to each cover slip for 1 h at 37 °C in a 5% CO_2_ (cell attachment). After incubation, neutrophils were stimulated with PMA (100 ng/mL), calcium ionophore (A23187) (100 nM), or LPS (100 ng/mL) at 37 °C and 5% CO_2_ for 2 h, washed twice with 1×PBS, fixed with 4.2% paraformaldehyde (PFA) for 15 min, and blocked with 2% bovine serum albumin (BSA). NETs were determined by the extracellular colocalization of anti-citrullinated histone H3 antibody at 1:200 dilution (Abcam, Cambridge, MA, USA) with the nuclear staining by 4′,6-diamidino-2-phenylindole (DAPI) at 1 µg/mL for 10 min. The secondary antibody was Alexa Fluor 488 at 1:400 dilution (Invitrogen, Carlsbad, CA, USA). Fluorescent images were obtained by a fluorescent microscope (Olympus, Hachioji, Tokyo, Japan), and the number of cells with NET formation was counted and reported per 100 cells. The phagocytic activity was performed by flow cytometry-based assay as mentioned below. 

### 4.4. Flow Cytometry Analysis

All flow cytometric analyses were measured by a FACS Canto II cytometer (BD Biosciences, Franklin Lakes, NJ, USA) with the FlowJo V10 (Ashland, Wilmington, DE, USA). Neutrophils and PBMCs (5 × 10^5^ cells) were suspended in staining buffer and labeled with antibody panels. Antibodies used in the experiments were fluorescein isothiocyanate (FITC)-conjugated anti-CD66b (BD Biosciences), phycoerythrin (PE)-conjugated anti-CD63 (BD Biosciences), PE-conjugated anti-arginase I (BioLegend, San Diego, CA, USA), PE-cyanine 5 (cy5)-conjugated anti-CD184 (BD Biosciences), allophycocyanin (APC)-conjugated anti-CD62L (BD Biosciences), and APC-cy7-conjugated anti-CD11b (BD Biosciences). The cells were labeled for 30 min at 4 °C in the dark with antibodies before being fixed with 4.2% paraformaldehyde (BD Biosciences). Analysis was gated by dot-plot analysis, and at least 25,000 cells were acquired per sample. To analyze phagocytic capacity, unstimulated and stimulated cells (2 × 10^5^ cells) were mixed with pHrodo *S. aureus* Bioparticle^TM^ (Thermo Fisher, Carlsbad, CA, USA) in 37 °C for 1 h before labeling with anti-CD66b antibody (a neutrophil biomarker) and being fixed with 4.2% paraformaldehyde. Phagocytosis was demonstrated through a percentage of CD66b-positive cells with positive bioparticles (emission: PE) and reported with mean fluorescent intensity (MFI). Apoptosis was quantified by annexin V-FITC and propidium iodide (PI) staining with early apoptotic cells (annexin V+/PI−) and late apoptosis (annexin V+/PI+) using an apoptosis assay (BD Biosciences) following the manufacturer’s instructions. The reactive oxygen species (ROS) production was assessed by detecting fluorescent alteration in the cells loaded with dihydroethidium (DHE) (Thermo Fisher, Waltham, MA, USA) following a previous protocol [11]. Briefly, the cells (2.5 × 10^5^ cells) were resuspended in complete RPMI-based media containing 2.5 μM DHE and incubated for 30 min at 37 °C in the dark. The cells were washed with cold D-PBS and resuspended in cold 4.2% paraformaldehyde and analyzed by flow cytometry using 485 nm (excitation; blue laser) and 520 nm (emission; PE) filters, gated by dot-plot analysis (10,000 cells were acquired per sample), and reported as MFI.

### 4.5. Human T Cell Isolation and Proliferation Assay

Isolated T cells in the suspension were captured by immunomagnetic negative selection using the EasySep™ Human T Cell Isolation Kit (Stem cell Technologies, Vancouver, BC, Canada), while a trypan blue assay (Thermo Fisher, Waltham, MA, USA) was used to detect cell viability. For cell proliferation, isolated CD3+ T cells (1.0 × 10^5^ cells) or T cells in PBMC (containing 1.0 × 10^5^ CD3+ T cells) were seeded into a 96-well plate and stimulated with CD3/CD28 dyna-bead (Thermo Fisher, Waltham, MA, USA) using the unstimulated T cells (no dyna-bead) as controls according to the manufacturer’s protocol. After 4 days, the cells were harvested and stained with APC-conjugated anti-CD3 antibody (BD Biosciences) and analyzed by flow cytometry LSR II (BD Biosciences) as previously described [16]. The proliferation rate of T cells from the PBMC or peripheral blood was assessed by carboxyfluorescein succinimidyl ester (CFSE) (Thermo Fisher, Waltham, MA, USA) dilution as previously described [16].

### 4.6. Wright Giemsa Stain Assay

The PBMC and neutrophils fractions after gradient separation were smeared, naturally dried, stained with the Wright–Giemsa solution (Merck, Darmstadt, Germany) at room temperature for 1–2 min, mixed with PBS (pH 7.2), and washed by sterile water. The results of cell staining were observed using a bright field microscope (Nikon, Shinagawa, Tokyo, Japan).

### 4.7. Statistical Analysis

GraphPad Prism 8.0 (GraphPad Software, Inc.) was used for statistical analysis and graph presentation. The normality test was performed by the two-sample Kolmogorov–Smirnov test using *p* ≥ 0.05 as the passing of the test. Student’s *t*-test, the Mann–Whitney U test and one-way analysis of variance (ANOVA) with Tukey’s analysis, or Kruskal–Wallis were used for the two- and three-group comparison, respectively. A *p*-value < 0.05 was considered a statistically significant difference. The correlation between the percentage of different neutrophil subpopulations and several neutrophil characteristics (chemotaxis, phagocytosis, ROS, NETs, and apoptosis) was calculated by Pearson’s or Spearman’s rank correlation.

## 5. Conclusions

In conclusion, iron overload in patients with β-thalassemia induced changes in neutrophil subpopulations (increased aged, immature, and low-density neutrophils), which might alter immune responses, partly through T cell suppression. Due to possible immune suppression caused by iron overload, effective iron chelation is encouraged. The understanding of neutrophil heterogeneity and functional plasticity in β-thalassemia may develop new therapeutic approaches in the future.

## Figures and Tables

**Figure 1 ijms-25-10651-f001:**
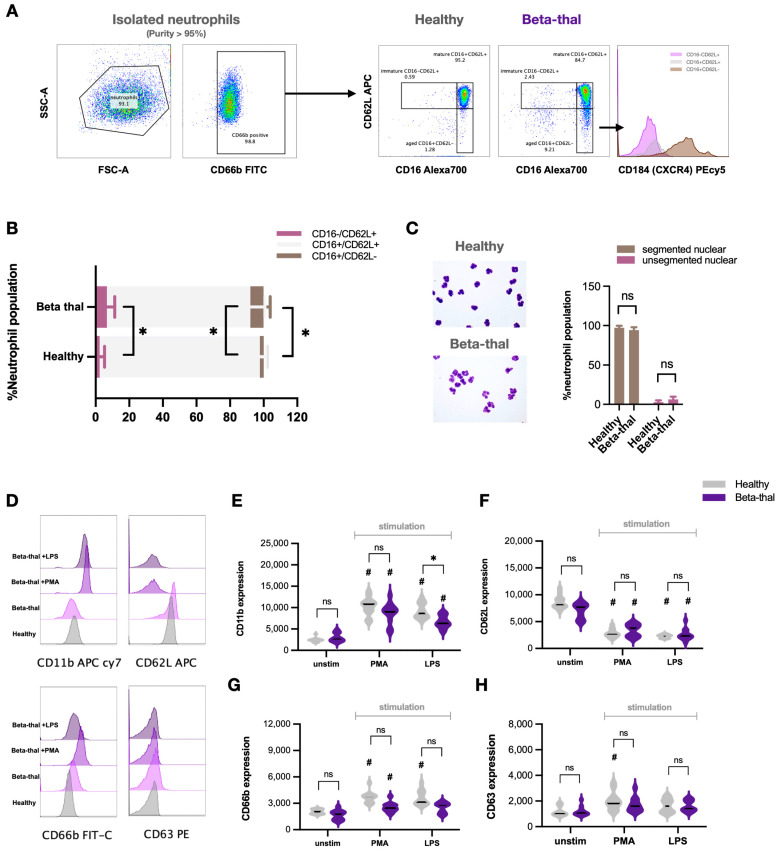
Elevated immature and aged neutrophils with some different characteristics in β-thalassemia compared with healthy controls. Scheme of the pattern of flow cytometry analysis started from neutrophil fraction (density gradient separation) (more than 95% of CD66b-positive cells) using CD16 and CD62L to separate into immature neutrophils (CD16− CD62L+), mature neutrophils (CD16+ CD62L+), and aged neutrophils (CD16+ CD62L−), and aged neutrophils were further tested by CD184 (a biomarker of CXCR4) (**A**). Abundance of mature neutrophils (light gray), immature neutrophils (dark gray), and aged neutrophils (purple color) in healthy controls and patients with β-thalassemia using flow cytometry analysis (**B**) together with an abundance of mature neutrophils (multi-segmented cells) and immature cells (band form) using conventional Wright’s stain (with the representative Wright’s stain pictures) (**C**) are demonstrated (*n* = 23 for patients with β-thalassemia and *n* = 20 for healthy control). For cell function analysis, the expression of CD11b (β-integrin), anti-CD62L (L-selectin), CD66b (carcinoembryonic antigen-related cell adhesion molecule 8 or CEACAM8), and CD63 (Tetraspanin) of neutrophils from healthy controls (gray color) and patients with β-thalassemia (purple color) after stimulation by phorbol myristate acetate (PMA; a neutrophil activator) or lipopolysaccharide (LPS; a potent pro-inflammatory stimulator) or unstimulated cells (unstim) with representative pictures of flow cytometry patterns (**D**–**H**) are demonstrated (*n* = 23/group for unstim, PMA, and LPS). *, *p* < 0.05 for figure B; #, *p* < 0.05 vs. unstim neutrophils from healthy controls; ns, non-significance.

**Figure 2 ijms-25-10651-f002:**
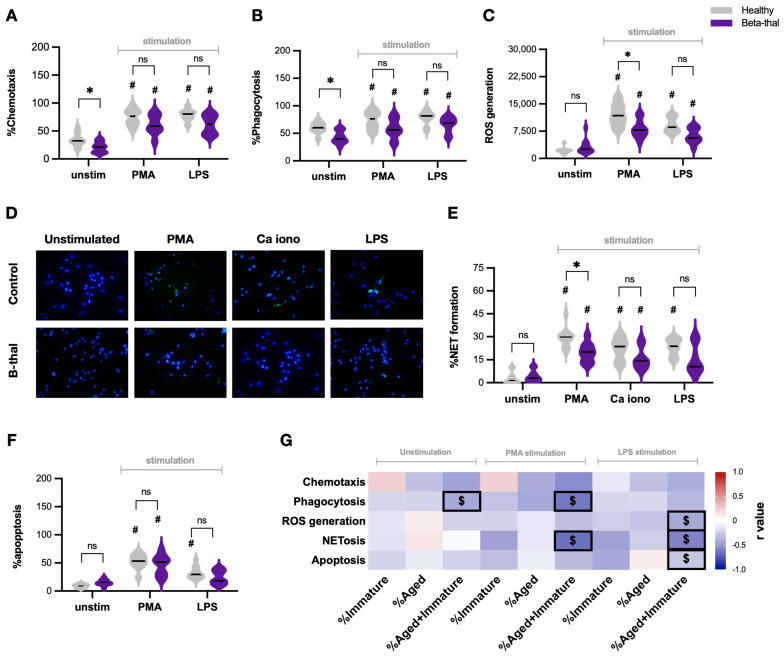
Different characteristics of β-thalassemic neutrophils compared with healthy neutrophils. Characteristics of neutrophils from patients with β-thalassemia (Beta-thal) or from healthy controls (Healthy) without stimulation (unstimulated cells) (unstim) or stimulation with phorbol myristate acetate (PMA; a neutrophil activator) or lipopolysaccharide (LPS; a potent pro-inflammatory stimulator) or calcium ionophore (Ca iono; an inducer for neutrophil extracellular traps (NETs)) as indicated by chemotaxis (**A**), phagocytosis (**B**), production of reactive oxygen species (ROS) (**C**), NET formation with representative immunofluorescent pictures using anti-citrullinated histone H3 (CitH3) (green color) and 4′,6-diamidino-2-phenylindole (DAPI; a nucleus staining color) (**D**,**E**), and apoptosis (**F**) are demonstrated (*n* = 23 for patients with β-thalassemia and *n* = 20 for healthy controls) (*, *p* < 0.05 for figure (**A**–**C**); #, *p* < 0.05 vs. unstim neutrophils from healthy controls; ns, non-significance). Heat map of the r-square (r^2^) values (coefficient of determination) calculated by Spearman’s rank correlation coefficient of β-thalassemic neutrophils (unstimulating or stimulation with PMA or LPS) between the x-axis of percentage of different neutrophils, including immature (CD16− CD62L+), aged (CD16+ CD62L−), and combined cells (immature plus aged cells), versus functional parameters in the y-axis to demonstrate directions of the responses (red and blue colors are positive and negative correlation, respectively, and the $ sign indicates significant correlation at *p* < 0.05) (**G**) are shown (*n =* 23 for immature or aged neutrophils and *n* = 46 for combined immature plus aged cells).

**Figure 3 ijms-25-10651-f003:**
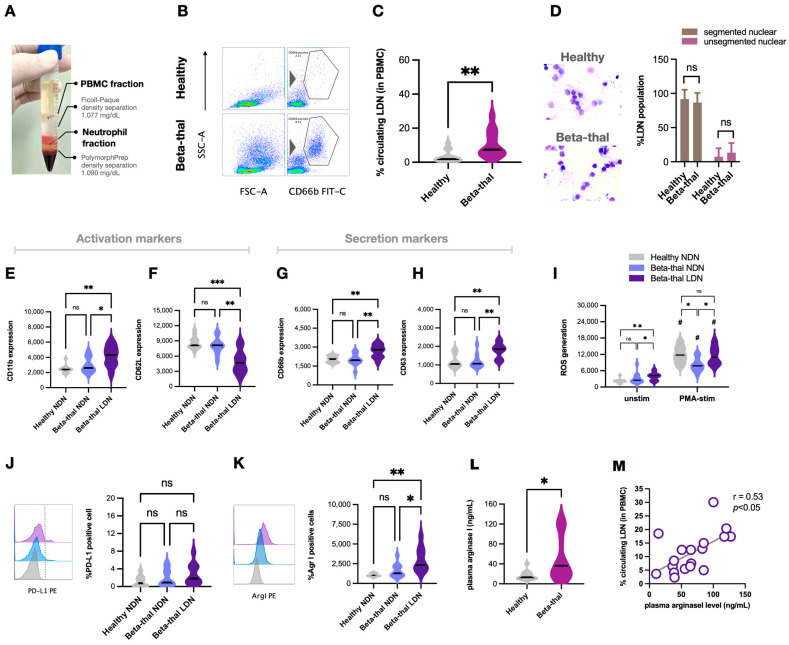
Elevated low-density neutrophils (LDNs) with some different characteristics of β-thalassemia LDNs compared with normal-density neutrophils (NDNs). A representative picture of density-gradient separation demonstrated the fractions of polymorphonuclear cells (PBMCs) and neutrophils at the upper part of Ficoll-paque and between Ficoll-paque and PolymorphPrep, respectively, (**A**) and the representative pictures of flow cytometry analysis from the PBMC fraction from patients with β-thalassemia (Beta-thal) and healthy volunteers (Healthy) (**B**) with an abundance of low-density neutrophils (LDNs; neutrophils in the PBMC fraction) using flow cytometry analysis (**C**), and Wright’s stain (conventional light microscope) with representative staining pictures (**D**) are demonstrated (*n =* 18–23/group). Flow cytometry analysis of regular normal density neutrophils (the neutrophil fraction of density gradient separation) from patients (Beta-thal NDN) and controls (healthy NDN) versus beta-thal LDNs as indicated by activation markers (CD11b and CD62L) (**E**,**F**), secretory markers (CD66b and CD63) (**G**,**H**), reactive oxygen species (ROS) production (an alteration in fluorescent signaling in dihydroethidium-loaded cells) with and without stimulation of phorbol myristate acetate (PMA) (**I**), PD-L1 (**J**), and arginase I expression (**K**) are demonstrated (*n =* 18–23/group). Levels of plasma arginase I from patients and healthy controls (**L**) together with the correlation between percentage of LDNs and plasma arginase level of patients (**M**) are also shown (*n =* 18–23/group). *, *p* < 0.05; **, *p* < 0.01; ***, *p* < 0.001; #, *p* < 0.05; ns, non-significance.

**Figure 4 ijms-25-10651-f004:**
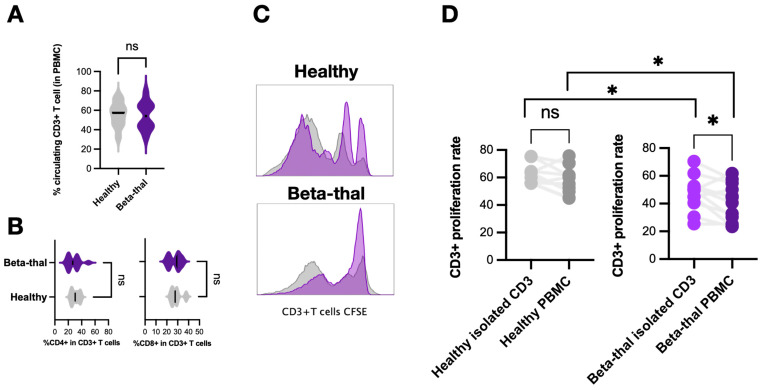
Reduced T cell proliferation in patients with β-thalassemia compared with healthy controls. The abundance of total peripheral blood CD3+ T cells, T helper (Th) cells (CD3+ CD4+), and cytotoxic (Tc) cells (CD3+ CD8+), with representative flow cytometry patterns from patients with β-thalassemia (Beta-thal) and healthy volunteers (Healthy) (**A**–**C**), is demonstrated (*n* = 16–20/group). Additionally, the CD3+ proliferation rate using carboxyfluorescein succinimidyl ester (CFSE) assay of CD3+ T cells isolated from the peripheral blood (isolated CD3) or peripheral blood mononuclear cell fraction (PBMC) from Beta-thal and Healthy groups (**D**) is also shown (*n* = 16–20/group). *, *p* < 0.05; ns, non-significance.

**Figure 5 ijms-25-10651-f005:**
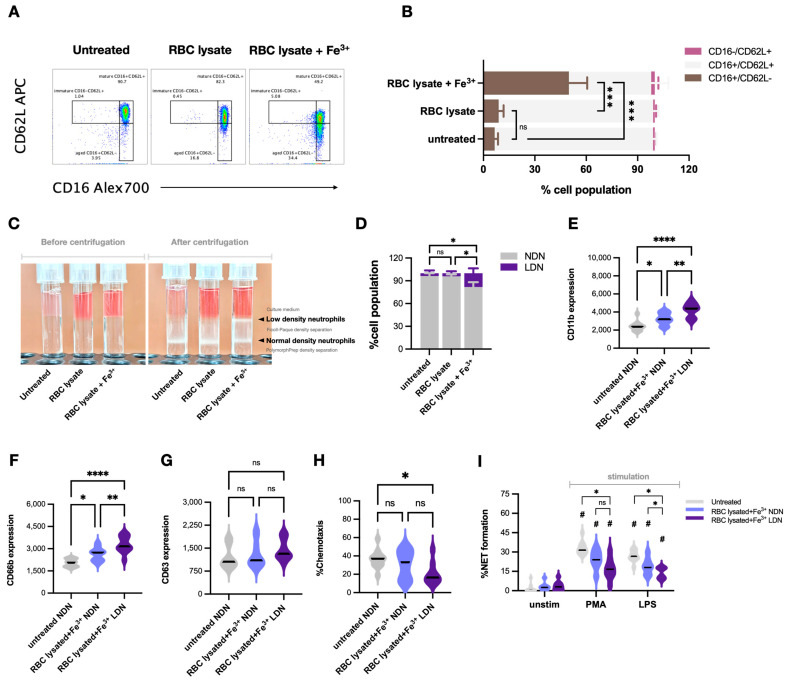
Induction of low-density neutrophils (LDNs) from normal-density neutrophils (NDNs) of healthy controls and differences between iron activation in LDNs and NDNs. Schema of flow cytometry analysis started from neutrophils from healthy volunteers (untreated) and the neutrophils incubated by red blood cells (RBCs) lysate alone or RBC lysate with ferric ions (Fe^3+^) (**A**) and abundance of mature neutrophils (CD16+ CD62L+), aged neutrophils (CD16+ CD62L−), and immature-liked neutrophils (CD16− CD62L+) (**B**) are demonstrated (the experimental group with ferric ions alone is not demonstrated due to similarity to the RBC lysate + Fe^3+^ group). The representative pictures of density gradient separation tubes with untreated neutrophils or neutrophils with RBC lysate alone or with ferric ions at the time before and after centrifugation to separate low-density neutrophils (LDNs) and normal-density neutrophils (NDNs) (**C**) with a graph presentation of the percentage of LDNs and NDNs in each experimental group (**D**) are also demonstrated. Function of untreated NDNs, NDNs with RBC lysate alone, or RBC lysate with ferric ions (RBC lysate+Fe^3+^) as determined by flow cytometry analysis using CD11b, CD66b, and CD63 (**E**–**G**), chemotaxis activity (**H**), and percentage of neutrophil extracellular traps (NETs) using these neutrophils with or without phorbol myristate acetate (PMA) or lipopolysaccharide (LPS) (**I**) are shown. The data were retrieved from 5 independent experiments. *, *p* < 0.05; **, *p* < 0.01; ***, *p* < 0.001; ****, *p* < 0.0001, #, *p* < 0.05 vs. untreated unstimulated neutrophils; ns, non-significance.

**Figure 6 ijms-25-10651-f006:**
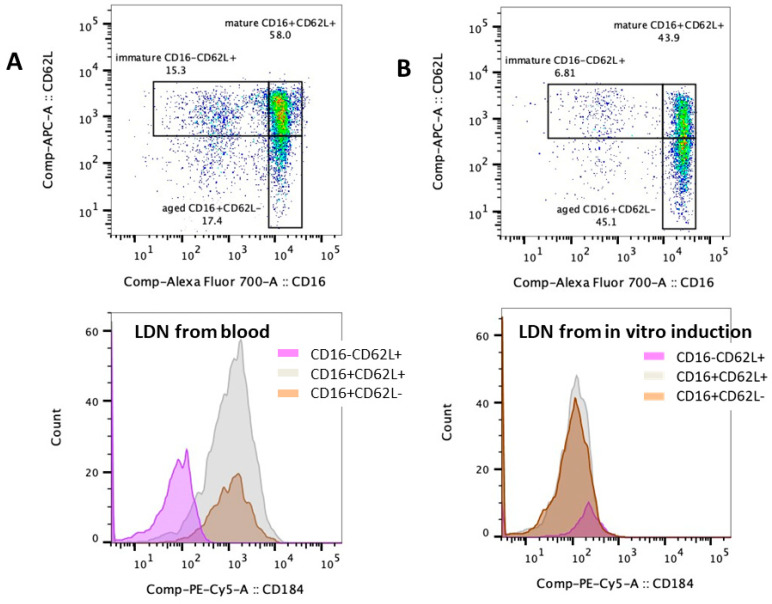
The differences between low-density neutrophils (LDNs) from patients with β-thalassemia and LDNs from in vitro induction. The representative pictures of flow cytometry analysis using antibodies against CD16 and CD62L (**upper part**) together with CD184 (**lower part**) from LDNs of patients (LDNs from blood) (**A**) and induced LDNs from normal-density neutrophils (NDNs) (**B**) using red blood cell lysate plus ferric ions (LDNs from in vitro induction) are demonstrated.

**Table 1 ijms-25-10651-t001:** Epidemiologic data of the participants.

	Patients (23) *	Control (20) *
Age (Year)	22 ± 3	22 ± 7
Gender (Female/Male)	14/18	12/8
Serum Endotoxin (EU/mL) #	0.07 ± 0.05	0
Splenectomy (%)	6/23 (26%)	None
Hemoglobin (g/dL) #	8.5 ± 0.6	13 ± 3
No. of Transfusions (6 Months before Recruitment)	4.5 ± 1.3	None
Lymphocyte Count (×10^3^/mL)	11.3 ± 6.5	13.1 ± 4.5
Platelet (×10^5^/mL)	2.2 ± 1.1	1.9 ± 1.5
Total Bilirubin (mg/dL) #	1.7 ± 0.5	0.6 ± 0.2
Direct Bilirubin (mg/dL)	0.6 ± 0.3	0.3 ± 0.1
Indirect Bilirubin (mg/dL)	1.1 ± 0.5	0.2 ± 0.1
Fasting Plasma Glucose (mg/dL)	92 ± 11	82 ± 17
Serum Calcium (mg/dL)	8.9 ± 0.5	9.2 ± 0.9
Serum Phosphate (mg/dL)	4.3 ± 0.3	4.2 ± 0.8
Serum Albumin (g/dL)	4.2 ± 0.5	4.5 ± 0.3
Serum Alanine Transaminase (ALT) (U/L) #	117 ± 25	22 ± 15
Serum Aspartate Transaminase (AST) (U/L) #	102 ± 35	30 ± 7
Blood Urea Nitrogen (BUN) (mg/dL)	22 ± 5	21 ± 4
Serum Creatinine (mg/dL)	1.2 ± 0.4	1.1 ± 0.3
Serum Ferritin (ng/mL)	3172 ± 1244	97 ± 27

Values are means ± standard error (SE); *, number of subjects; #, *p* < 0.05.

## Data Availability

The original contributions presented in the study are included in the article; further inquiries can be directed to the corresponding author.

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
