# Peer review of "Neutrophil Diversity (Immature, Aged, and Low-Density Neutrophils) and Functional Plasticity: Possible Impacts of Iron Overload in β-Thalassemia"

_ijms, 2024, doi:10.3390/ijms251910651_

Round 1
Reviewer 1 Report
Comments and Suggestions for Authors
The submitted manuscript presents the results of the original research, focused on establishing the neutrophil diversity and their functional plasticity in patients with β-thalassemia, in comparison with the control group (healthy volunteers). The manuscript is of rather high quality and within the scope of IJMS. Nevertheless, it should also be revised, following the comments listed below.
Major comments:
The abstract is significantly too long. It should be limited to maximum 200 words.
The first sentence of introduction should briefly remind the reader what β-globin is and what is its role.
Line 580, what test has been used to check the normal distribution?
Why the Authors have divided the conclusions into two sections, one starting in Line 465 and another starting in Line 588? They should be merged.
Line 594, do you mean Fe chelation as a therapy? While I do agree, this topic should also be discussed in the introduction.
Table 1, was the level of Hemoglobin really 13 in all of the subjects (control) leading to SD=0? It is very unlikely and should be double checked.
Lines 460-462, please list those experiments and indicate the future directions.
While the Authors mention the role of iron multiple times, the level of iron has not been measured in the samples. Why?
Figure 5, what’s the meaning of three asterisks (“***”) i.e. in the (B), this has not been described in the Figure’s caption.
Minor issues:
Line 106, it should be “Results”
Line 137, it should be “Scheme”
Line 473, it should be “Materials and Methods”
Line 330, and other places too, it should be “Fe3+”
Line 465, similar, it should be “Conclusions” (plural)
Line 503, it should be “CO2”
Comments on the Quality of English LanguageWhile the English is quite OK, there are some grammar and style mistakes, i.e. in lines 49-50 “Several studies in β-thalassemia have demonstrated defects in the immune system in β-thalassemia”.
Reviewer 2 Report
Comments and Suggestions for Authors
The manuscript concerns neutrophil subpopulations in beta-thalassemia and the impact of iron overload. The topic is clinically relevant, and a few reports are in the literature.
The article's aim is clear and well-defined. The methodology is adequate; however, the abstract and part of some Figures need to be modified. Also, two conclusions should be partially rewritten. There are several questions based on iron content in patients' serum and ferritin levels. Figure 1, part C, should be changed; the colors on the bars are not adequate to distinguish segmented from unsegmented neutrophils. Why the other IgG Fc receptors were not analyzed in these neutrophils? Why was phagocytosis, which has been described to be impaired in these patients, not performed? The decrease in response to PMA and LPS has to be discussed further, as well as the difference in cell density, which may also be due to a reduction in cell granules. How is the expression of CD15 and CD32 in low and normal-density neutrophils? In the rationale, Figure 3 suggests that low-density neutrophils are more responsive than normal density, part I, why do they express more arginase? Figure 4 is unconvincing; T-cell proliferation is independent from low-density neutrophils. Most probably, the intracellular iron content is responsible for changes see doi: 10.4049/jimmunol.1901399
Figure 5 contradicts the previous results since it shows that the cells are less response and express CD63 receptors. Why? The netosis protocol is not well defined, and the results should be better defined. Figure 6 is confusing, the role of CD184 is not well defined. What is the response of these neutrophils to chemokines? Is spherocytosis a vital phenomenon in thalassemia? The authors should discuss it further
Finally, several items must be considered. The abstract should be 200 to a maximum of 250 words and informative; please modify it. The discussion should be rewritten based on changes in cell viability. Will the results differ if iron chelators were used in vitro?
Comments on the Quality of English LanguageThere are several sentences that need revision as well as some minor grammatical mistakes
Round 2
Reviewer 1 Report
Comments and Suggestions for Authors
The Authors have revised and improved their work. Current version can be accepted.
Reviewer 2 Report
Comments and Suggestions for Authors
Even though the authors did not comply with all the requests I solicited, the article can be accepted considering it is a preliminary report. I encourage the authors to perform more suitable assays to analyze the differences in the two populations further.
Comments on the Quality of English LanguageThere are several grammatical mistakes in the text.